# Effect of Enrichment Items on the Physiology and Behavior of Sows in the Third Trimester of Pregnancy

**DOI:** 10.3390/ani12111355

**Published:** 2022-05-26

**Authors:** Shuangshuang Li, Hongqing Hu, Jian Huang, Yuxuan Yang, Weijing Xu, Junfeng Chen, Jiawei Wan, Lianghua Li, Rong Zheng, Siwen Jiang, Jin Chai

**Affiliations:** 1Agricultural Ministry Key Laboratory of Swine Breeding and Genetics & Key Laboratory of Agricultural Animal Genetics, Breeding, and Reproduction of Ministry of Education, Huazhong Agricultural University, Wuhan 430070, China; lishuangshuang188@163.com (S.L.); hongqing.hu@doct.uliege.be (H.H.); huangjian978543177@163.com (J.H.); yangyuxuan261@163.com (Y.Y.); q2385685069@163.com (W.X.); wjwgo950428@163.com (J.W.); zhengrong@mail.hzau.edu.cn (R.Z.); jiangsiwen@mail.hzau.edu.cn (S.J.); 2The Cooperative Innovation Center for Sustainable Pig Production, Wuhan 430070, China; 3Henan Key Laboratory of Farm Animal Breeding and Nutritional Regulation, Institute of Animal Husbandry and Veterinary Science, Henan Academy of Agricultural Sciences, Zhengzhou 450002, China; afeng008@163.com; 4Institute of Animal Husbandry and Veterinary Medicine, Hubei Academy of Agricultural Sciences, Wuhan 430064, China; lilianghua988@163.com

**Keywords:** confinement stalls, environmental enrichment, behavior, physiology, reproductive

## Abstract

**Simple Summary:**

Environmental enrichment can mitigate the damage to animal welfare induced by modern intensive pig farming practices, especially when pregnant sows are raised in confined stalls. This study investigated the effects of enrichment items (such as adding pine and scented wood in confinement stalls) on the physiology and behavior of sows during late gestation. Adding both pine and scented wood in confinement stalls was shown to alleviate chronic stress and stereotypical behavior of sows, suggesting their potential as an interesting and feasible way to reduce welfare compromise.

**Abstract:**

Modern intensive pig breeding harms animal welfare, which is especially noticeable for pregnant sows kept in confinement stalls. This study aimed to evaluate the effects of enrichment items on the movement and physiological parameters of sows in the third trimester of pregnancy. A total of 30 large white pregnant sows were randomly divided into three equal treatment groups (n = 10): control, pine wood, and scented wood groups. Interestingly, compared with the control group, the sows in the pine wood or scented wood groups showed less ventral lying and more lateral lying behavior (*p* < 0.01), coupled with significant reduction in the frequency of scratching and sham-chewing (*p* < 0.01), but with no significant difference in the degree of preference for these enrichment items (*p* > 0.05). Additionally, the sows in the pine wood or scented wood groups also decreased significantly in the concentration of immunoglobulin A (IgA) (*p* < 0.01) and the concentration of tumor necrosis factor-α (TNF-α) (*p* < 0.05) throughout the late pregnancy period. Overall, adding enrichment items to confinement stalls can alleviate the chronic stress and the stereotypic behavior of sows, suggesting their potential to reduce welfare compromise.

## 1. Introduction

Confinement stalls are significant features of pig intensive production, which improved the utilization efficiency of the livestock house and reduced individual production cost. However, compared with group pens, sows raised in confinement stalls showed more stereotypic behaviors [1]. When the natural behavior of pigs is limited and their instincts are not satisfied, it may impair animal welfare and lead to a series of abnormal phenomena, such as chronic stress, stereotypic behavior (sham-chewing, biting, sitting), lameness, and the risk of decubitus ulcer [2,3,4].

The European Commission Directive (2008/120/EC) indicates that pigs must have permanent access to sufficient quantities of materials for appropriate exploration and operation, such as straw, hay, wood, sawdust, mushroom compost, peat or mixtures of these materials, with the purpose to increase animal welfare. Environmental enrichment can reduce the time of inactivity and harmful social and aggressive behaviors of pigs, while increasing their exploration time can also reduce their stress and promote their productive performance [5,6,7]. The use of aromatized objects was reported to reduce the duration of agonistic behaviors of the piglets and the exchange of fragrances could increase the interest of the animals in the new object [8], reinforcing the notion that piglets alter their behaviors according to the environmental stimuli they receive. Aside from behavioral indices, physiological indicators are also often used as a measure of stress hormones to assess animal welfare [9]. Cortisol and IgA in saliva have become a commonly used marker for pig stress evaluation [10,11,12]. With the extension of space restriction period of pregnant sows (1–5 parities), the concentrations of cortisol, IgA and IL-6 in serum of pregnant sows increased significantly, inferring that space restriction has a serious impact on the psychological status of sows [13].

The third trimester is a relatively important period of pregnancy, because the growth of the fetus during pregnancy is slow in the first trimester, fast in the second trimester, and faster in the third trimester, forming 1/3 of the fetal body weight in the third trimester [14]. Appropriate housing for pregnant pigs in the second to third trimester may assist them to receive adequate nutrition for the growth of the placenta, fetus, and maternal body. Confinement stalls can give an appropriate feed intake according to the situation of each pregnant sow, so as to avoid fighting with each other and ensure sufficient energy [15]. However, there is no relevant research on the effect of adding enrichment items on the performance of sows in the third trimester of pregnancy.

Therefore, the purpose of this study was to evaluate the effects of enrichment items (pine wood and scented wood) on the behavior, physiology, and reproductive performance of sows in the third trimester of pregnancy, as well as to determine whether there are differences between the two enrichment items in their effects. We hope that this study can provide a reference for reducing welfare compromise of pregnant sows in confinement stalls and screening out enrichment items that can increase production efficiency.

## 2. Materials and Methods

### 2.1. Animals, Facilities, and Management

In this study, 30 French genetically pure large white pregnant sows (12–15 weeks of gestation) with the same genetic background, (third) parity and expected delivery date (3 ± 1.5 d) were randomly divided into three groups (n = 10): the control without any enrichment item, and two treatment groups (fresh pine wood or scented wood).

During the trial, sows were kept in separate stalls (2.1 m × 0.6 m) adjacent to each other without contact and equipped with slatted floors, duck-billed drinking fountains and communal feeding troughs (Figure 1). Gestating sows were fed full-price granule compound feed (the main ingredients are shown in Table 1), and each group was fed twice a day (8:00 a.m. and 15:00 p.m.) with the feeding amount of 2.3–2.4 kg/head/day. During the experiment, all the sows were inspected regularly every day, and the piggery was disinfected once a week on Friday. Meanwhile, the temperature (27 ± 3 °C) and humidity (60%) were monitored, and no sows were immediately removed from the experiment because of returning to estrus or abortion.

### 2.2. Enrichment Items and Treatment Schedule

Pine: pine made of fresh pine wood with a diameter of 6–8 cm and a length of 0.5 m.

Scented wood: scented wood made of dry, odorless ordinary wood, with a diameter of 6–8 cm and a length of 0.5 m, coupled with 2 holes on each piece of wood to store agar, pine essence and pigment, and they were tied around the stalls of sows at 12 weeks of gestation, as shown in Figure 2.

### 2.3. Behavior Recording

During the experiment, all activities of the sows were filmed and documented from 6:00 a.m. to 18:00 p.m. using a digital video system (Hikvision DS-ITS, Hangzhou Hikvision Co., Ltd., Hangzhou, China). In order to avoid the influence of feeding time and hot temperature at noon, we selected four time periods (7:00–8:00, 9:00–10:00, 14:00–15:00 and 16:00–17:00) every day for observation and recording. Instantaneous scan sampling was performed for behavioral postures (standing, ventral lying, lateral lying, sitting), with recording being paused every 5 min, and 48 events completed every day (240 min of video recording per day). A frequency chart of postural behaviors was established by recording the number of postural behavior events in each pig in 48 observed events per day to calculate the percentage of postural behaviors per pig per day throughout the trial period. For other behaviors (sham-chewing, drinking, manipulation), the frequency and time of occurrence were recorded during the observation period of 4 h per day.

Sows in all confinement stalls were assessed individually and the number of sows performing each behavior was recorded and transferred to a spreadsheet. Behaviors are determined as defined in Table 2.

### 2.4. Reproductive Performance Measurement

The trial was completed at the beginning of the 16th week of gestation and sows were transferred to farrowing houses. The effects of different enrichment items on the reproductive performance of pregnant sows were evaluated in terms of total number of piglets, number of live births, birth weight, number of stillbirths, number of mummies, and duration of farrowing. The gestating sows were scored for their labor as described in Table 3.

### 2.5. Saliva Collection

Sow saliva was collected at 7:00–9:00 a.m. every Wednesday for four weeks, using a self-made saliva collector (composed of medical gauze, medical degreased cotton, high-temperature sterilized disposable chopsticks and rubber bands), one sample per pig at a time. During sampling, the hand-held saliva collector was extended into the confinement stalls for the sow to chew freely. Once the cotton ball was wet, the saliva was collected in a centrifuge tube, followed by centrifugation for 10–15 min (3000 rpm) and storing the supernatant at −80 °C for further analysis. The ELISA kits (96T L180801256, L180710808, L190111127, Wuhan, China) were used to determine the contents (ng/mL) of cortisol, immunoglobulin A, and tumor necrosis factor-α in saliva, as instructed by the manufacturer of the kit. Briefly, after the sample was diluted 1000 times (to determine the TNF-α index in the original solution), the capture antibodies to the cortisol, IgA, and TNF-α were coated to the wells of 96-well ELISA plates, followed by incubating the samples with the kit solution and five washes in PBST to remove the nonspecific binding. Finally, the detection antibodies conjugated with horseradish peroxidase were incubated in the wells, and absorbance was measured at 450 nm using an ELISA reader. A standard curve was prepared with the concentration and absorbance of standards to produce a linear equation for quantifying the experimental samples (Appendix A).

### 2.6. Data Analysis

The data were processed with EXCEL 2016 and SPSS 22.0 software. GraphPad Prism 8.0 was used for graphing. The difference between two groups was evaluated by paired *t*-test, while the comparison between the three groups was conducted by one-way ANOVA and the Duncan method for multiple comparison. All data are normally distributed. The results are presented as mean ± standard error of the mean (SEM). Unless otherwise stated, differences among groups were considered statistically significant at *p* < 0.05 and extremely significant at *p* < 0.01.

## 3. Results

### 3.1. Effect of Enrichment Items on Posture Behavior

Compared with the control group (with no access to enrichment items), the sows in the scented wood and pine wood groups showed more lateral lying and less ventral lying behavior, but no difference from the control group in standing and sitting behaviors. Meanwhile, the two enrichment item groups exhibited no significant difference in the above four behaviors (Figure 3). Lateral lying was shown to be a better way for pregnant sows to rest than ventral lying [19,20].

### 3.2. Effect of Enrichment Items on Ingestion and Stereotypic Behavior

In Figure 4A, the frequency of water drinking was shown to decrease in the pine group relative to the control. Then, we studied the influence of pine and scented wood on the stereotypic behavior by testing the duration and frequency of sham-chewing and scratching. In Figure 4B,C, the duration and frequency of both behaviors were seen to significantly decrease in the two enrichment item groups relative to the control, indicating that enrichment items are beneficial to reduce the stereotypic behavior of sows in the third trimester of pregnancy in the confinement stalls.

### 3.3. Effect of Enrichment Items on Manipulation Behavior

In this experiment, enrichment items were used to modify the environmental enrichment, so it is necessary to investigate whether the pregnant sows had preference for enrichment items. However, no significant difference was observed between the two groups of sows in their preference for pine or scented wood in the third trimester of pregnancy (Figure 5).

### 3.4. Effect of Enrichment Items on Cortisol, IgA and TNF-α in Saliva

In order to further explore the effect of enrichment items on the chronic stress behavior of pregnant sows, we investigated the concentration changes of cortisol, IgA, and TNF-α in the saliva of sows in the two enrichment item groups and the control group. In Figure 6A, except for the 12th week, there was no significant difference in the concentration of cortisol, but the two enrichment item groups were significantly lower than the control group in the concentration of IgA and TNF-α in the saliva of the sows throughout the third trimester of pregnancy (Figure 6B,C).

### 3.5. Effect of Enrichment Items on Reproductive Performance of Sows

As shown in Table 4, there were no significant differences between the two enrichment item groups and the control group in total number of piglets, number of live births, number of stillbirths, and number of mummies.

## 4. Discussion

The present study evaluated the effects of two different enrichment items (pine wood, and scented wood) on the behavior, physiology, and reproductive performance of large white sows in the third trimester of pregnancy, with the purpose of providing some feasible solutions for reducing the stress of pregnant sows and improving their welfare and reproductive efficiency. It was found that the addition of enrichment items improved the sows’ stereotypic behavior and chronic stress, but it had no significant effect on the reproductive performance of the sows.

### 4.1. The Effect of Enrichment Items on the Behavior of Pregnant Sows

Behavior can be used as part of animal welfare assessment due to its intuitive manifestation of the health state of animals. Long-term breeding of sows in monotonous confinement stalls significantly increases stereotypic behaviors (biting, sham chewing, and drinking) and causes severe psychological and physiological stress, thus harming their welfare [21,22]. In addition, scratching can lead to skin damage, which in serious cases can lead to systemic diseases and common symptoms of skin diseases, which are also considered to be harmful to the healthy growth of animals [23,24,25]. The results of our study showed that the frequency of sham-chewing and scratching of sows decreased significantly after pine and scented wood were added, and sows with pine wood enrichment items also drank less water. This was consistent with a previous report that the movement of pigs from barren pens to environmentally enriched ones tends to increase their time to explore, chew and play, thereby reducing their aggressive behaviors [26].

Lateral lying is not only an important aspect of maternal ability, but also important for the care and safety of piglets [27]. Compared to lateral lying, ventral lying is a relatively vigilant and active behavior, and lateral lying is probably the most ideal resting method for sows [19,20]. The present study found that after adding pine and scented wood, the frequency of lateral lying increased and the welfare compromise of sows in the late gestation period in the confinement stalls was reduced, which may be more helpful to protect the piglets. Meanwhile, the three groups showed no difference in dog sitting behavior. Previous studies have shown a positive correlation between the number of piglets crushed by the sow and the sitting time of the sow, suggesting the necessity to reduce the sitting behavior of sows during pregnancy and increase the frequency of their lateral lying [28].

Collectively, we can conclude that using pine and scented wood as enrichment items can reduce the stereotypic behaviors of sows in late pregnancy in the confinement stalls. However, with the addition of enrichment items, despite the higher likeliness of the sows to show lateral lying, the frequency of their sitting did not decrease significantly. The specific reasons need to be further verified.

### 4.2. Welfare Enrichment Items Selection Preference for Sows in the Third Trimester of Pregnancy

Environmental enrichment is aimed to change the barren captive environment and improve the biological functions of animals [29]. Van de Weerd [30] summarized the environmental enrichment in indoor production systems for piglets and mentioned the four characteristics for good environmentally-enriched materials: (1) increase of species behavior; (2) health maintenance and improvement; (3) increase of economic benefits; (4) applicableness in actual production.

In the present study, enrichment items (pine and scented wood) could significantly reduce the stereotypical behavior and physiological stress in sows, but with no significant difference between the two enrichment items in their effects. This is consistent with previous reports that fresh wood could reduce tail biting, ear biting and manipulation behavior, and the use of natural flavors in objects suitable for nibbling can encourage weaned piglets to maintain a longer strong interest [8,31]. Based on the frequency and duration of enrichment items manipulation by pregnant sows, there is no difference in the degree of interest in the two enrichment items (pine and scented wood) for the sows in the third trimester of pregnancy in the confinement stalls, suggesting that artificial scented wood has the same effect of pine wood. Scented wood is cheaper and more durable than pine, indicating that artificial scented wood can be used as an enrichment item to replace pine to achieve a more satisfactory result in practical applications.

### 4.3. The Effect of Enrichment Items on the Physiological Indexes in Saliva of Sows

When discussing animal welfare, a more important issue is not acute stress, but chronic stress, a situation where animals cannot adapt to their environment [32]. Munsterhjelm et al. [33] emphasized the association of a barren environment with signs of chronic stress.

Cortisol levels are a sign of stress, which tends to increase throughout pregnancy, reaching its maximal level in late gestation and before delivery, thus imposing a great risk on the survival of the embryo. In a barren environment, pigs show higher salivary cortisol levels than those in a straw-rich environment, indicating that animals in a monotonous environment can be stressed for a long time and may develop depression [30]. Exposure, especially long-term exposure to stress, is shown to impair immune function. Ciepielewski et al. [34] reported higher levels of TNF-α in pigs under long-term restraint. In 2010, IgA was identified as a non-invasive stress marker for pigs, whose IgA concentration in pig saliva was shown to decrease with the decrease of stress [12]. In the stress research of sows in confinement stalls in the third trimester of pregnancy, a single trial is clearly insufficient, and saliva biomarkers vary in response to different types of stressors, indicating the necessity to use different stimulus markers in stress studies [35]. Therefore, in this study, three physiological indicators of saliva (Cortisol, IgA, and TNF-α) were evaluated to comprehensively reflect the changes of stress level in pigs.

The results of this study showed that different enrichment items had no effect on the cortisol concentration in saliva of sows during the third trimester of pregnancy, but the pine wood and scented wood groups were lower than the control group in cortisol concentration. This is consistent with the research results on the effects of stocking density and environmental enrichment on pig behavior and fecal corticosteroid levels under commercial farm conditions. In barren and enriched pig pens, there is no difference in corticosteroid concentration in the feces of growing pigs, but the level of cortisol in enriched pens was lower than that in barren pens. [36]. This may be because physiological stress is affected by the reproductive stage [37]. Throughout the late stages of pregnancy, the concentrations of IgA and TNF-α were significantly reduced in the two enrichment items groups, indicating that enrichment items can reduce stress and improve the welfare of pregnant sows to some extent.

### 4.4. The Effect of Enrichment Items on the Reproductive Performance of Sows

In confinement stalls, due to space constraint, sows are unable to meet their basic needs to express species-specific behaviors such as exploration and communication, which will significantly increase their lying time and reduce standing behavior. Improper lying down was reported to compress the piglets in the abdomen of the pregnant sows and increase their mortality [38].

When the pig is not satisfied with certain behaviors, their behavior and production efficiency will be affected, causing welfare problems. Exploring and finding food is an inherent behavior of pigs at all stages of production, and the preparation of a nest is the only need for sows before delivery. In intensive farming units, providing environmentally-rich materials is one way to enable animals to express more of their innate behaviors, especially in the case of breeding, and avoid stress-induced delivery, which can cause birth defects in piglets [39]. Previous studies have shown no significant difference between the enrichment item groups and the control group in reproductive performance [40].

Consistent with previous studies, the present study showed no significant difference among the three groups in total number of piglets, live births, stillbirths, and mummies. Although they could improve the welfare of sows, pine wood and scented wood showed no effect on their reproductive performance, because the reproductive performance is also affected by genetic factors, management conditions, environmental factors, and nutrition, while enrichment items are only a small part of environmental factors, which may partially explain the lack of significant impact of enrichment items on reproductive performance.

## 5. Conclusions

This exploratory study showed that the provision of pine and scented wood can reduce the stereotypic behavior of sows in confinement stalls in the third trimester of pregnancy by increasing the complexity of the environment and providing channels for exploratory behavior, further reducing welfare compromise. Moreover, scented wood has the same effect as pine wood. Our study suggests that adding scented wood to the pig farm can be a potentially interesting and feasible way to reduce welfare compromise for sows.

## Figures and Tables

**Figure 1 animals-12-01355-f001:**
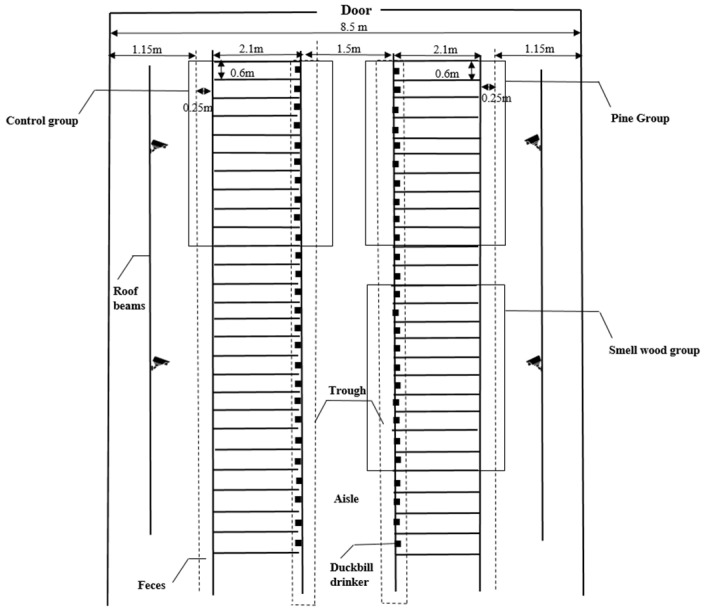
Schematic diagram of confinement stalls.

**Figure 2 animals-12-01355-f002:**
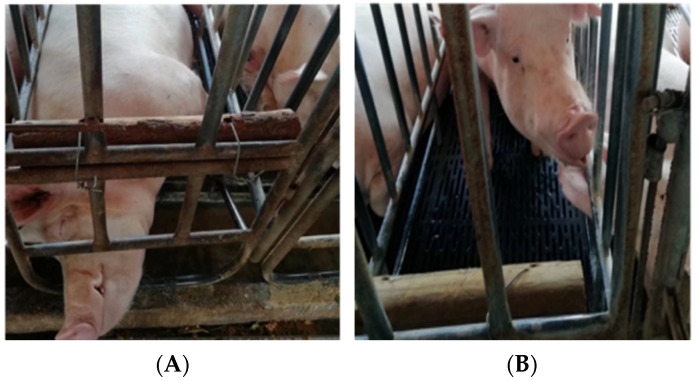
Enrichment items for pregnant sows in confinement stalls: (**A**) pine and (**B**) scented wood.

**Figure 3 animals-12-01355-f003:**
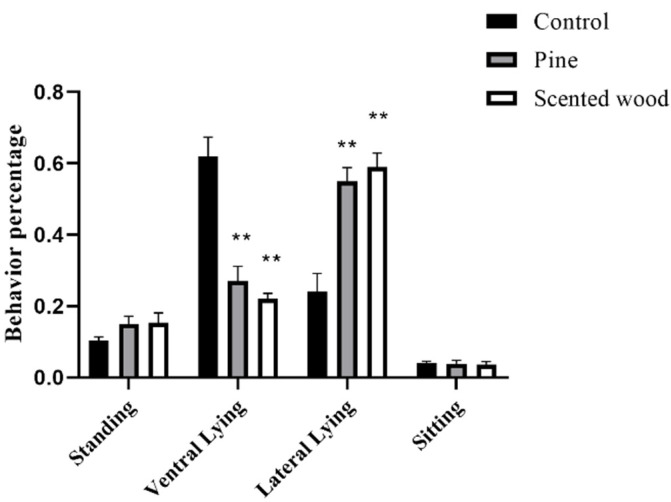
The effect of enrichment items on the posture behaviors of sows. ** *p* < 0.01 in enrichment item groups vs. control group. The data are expressed as mean ± SEM (n = 10).

**Figure 4 animals-12-01355-f004:**
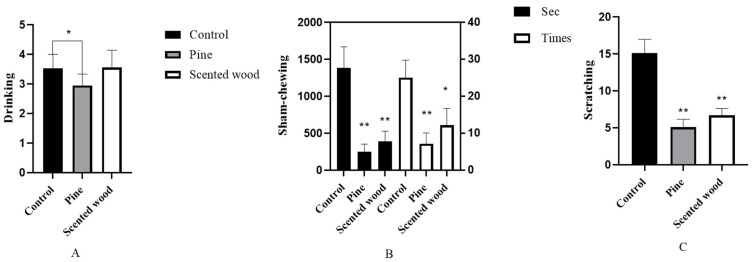
(**A**) The frequency of water drinking. (**B**,**C**) The frequency of sham-chewing and scratching behaviors. * *p* < 0.05 and ** *p* < 0.01 in enrichment item groups vs. control group. The data are expressed as mean ± SEM (n = 10).

**Figure 5 animals-12-01355-f005:**
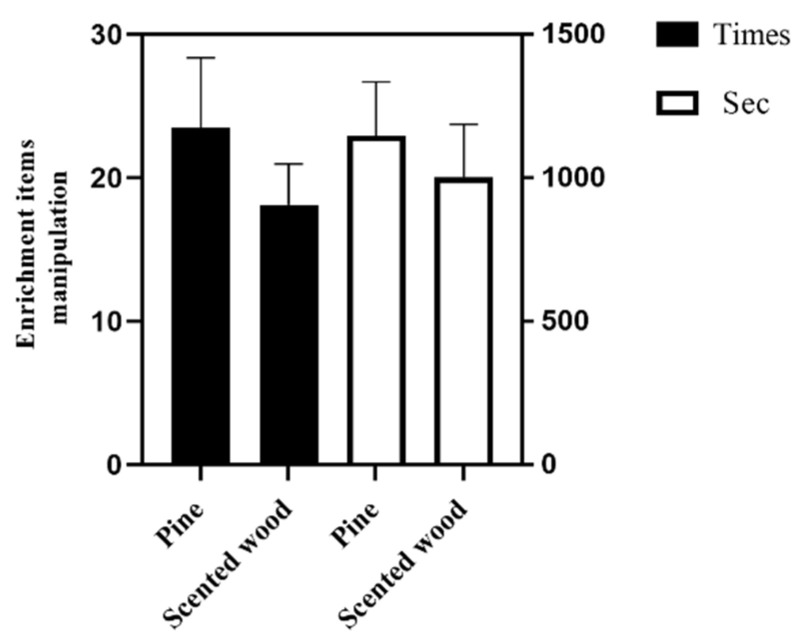
The manipulation frequency and duration of enrichment items. The data are expressed as mean ± SEM (n = 10).

**Figure 6 animals-12-01355-f006:**
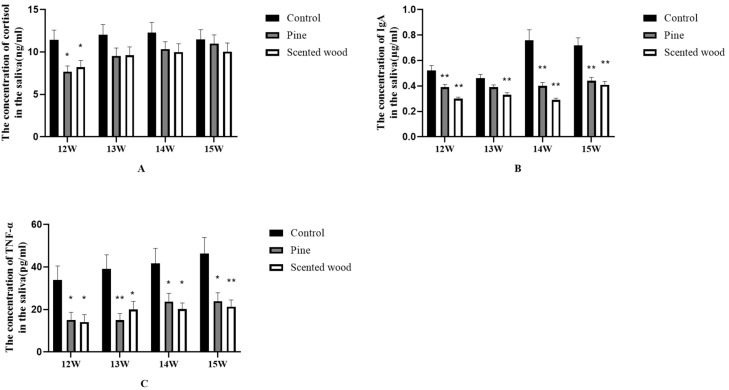
(**A**–**C**): Effect of enrichment items on the concentration of cortisol, IgA, and TNF-α in the saliva of sows (12 W–15 W). * *p* < 0.05 and ** *p* < 0.01 in the enrichment item groups vs. the control group. The data are expressed as mean ± SEM (n = 10).

**Table 1 animals-12-01355-t001:** Nutritional composition of pregnant sow feed.

Nutrient	Amount (%)
Crude protein (%)	15.00
Crude fiber (%)	8.00
Crude ash (%)	10.00
Calcium (%)	1.00
Total phosphorus (%)	0.40
Lysine (%)	0.85

**Table 2 animals-12-01355-t002:** Ethogram of behaviors used in the study.

Behavior	Description
**A. Posture**	
Standing	Supporting weight on the 4 limbs at the same time, but the forelimbs can kneel on the ground, and the limbs cannot be in the same plane
Lateral Lying	Lying with one shoulder on the floor, with 4 visible limbs [16]
Ventral Lying	Sow’s chest and abdomen touching the floor, with front legs stretched or folded under the body [17]
Sitting	Dog-sitting, with rear and front hooves on the floor [16]
**B** **. Ingestion**	
Drinking	Manipulating the drinker or apparently ingesting water
**C** **. Stereotypic behavior**	
Sham chewing	No food in the mouth and just oral activities with saliva
Scratching	Limb movements, licking, biting and rubbing [18]
**D. Enrichment interaction**	
Enrichment items manipulation	Arching, sniffing, chewing, and licking enrichment items

**Table 3 animals-12-01355-t003:** Labor score for pregnant sows.

Scores	Description
1	Delivery time is 1–1.5 h, and the delivery interval is 15–20 min/head; vulva is red and swollen; breathing is slightly deepened, lying calmly on her side and slightly humming.
2	The delivery time is 1.5–3 h, and the delivery interval is 20–30 min/head; the vulva is slightly edema; the birth canal shows slight hemorrhage; the breathing is fast; she likes to move and hums from time to time.
3	The delivery time is 3–4.5 h; the delivery interval is more than 30 min/head; the vulva has severe edema, hematoma, and a small amount of bleeding, coupled with abnormal breathing, small swings, and slight shouting.
4	The delivery time is 4.5–6 h and the delivery interval is 30 min/head; the vagina shows malignant edema, hematoma, and heavy bleeding, coupled with abnormal breathing, moderate swing, and violent shouting.
5	The delivery time is more than 6 h and the delivery interval is more than 30 min/head; the vagina shows malignant edema, hematoma, and heavy bleeding, coupled with abnormal breathing, large swing, and violent shouting.

**Table 4 animals-12-01355-t004:** Effect of enrichment items on the reproductive performance of late pregnant sows.

	Control	Pine	Scented Wood
	n = 10	n = 10	n = 10
Number of litters	13.11 ± 1.87	13.33 ± 1.05	12.90 ± 1.37
Number of live pigs	11.67 ± 1.25	12.33 ± 1.03	11.80 ± 1.14
Number of stillbirths	0.89 ± 0.45	1.00 ± 0.67	1.10 ± 0.43
Number of mummies	0.56 ± 0.44	0	0
Birth weight	15.20 ± 1.19	17.14 ± 1.39	16.30 ± 1.28
Average birth weight	1.36 ± 0.08	1.41 ± 0.07	1.43 ± 0.09
Labor score	2.78 ± 0.36	2.67 ± 0.24	2.40 ± 0.27
Duration of farrowing	227.22 ± 23.02	223.89 ± 11.72	214.00 ± 14.70

Note: The data are presented as mean ± SEM (n = 10).

## Data Availability

No new data were created or analyzed in this study. Data sharing is not applicable to this article.

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
