# Peer review of "Effect of Enrichment Items on the Physiology and Behavior of Sows in the Third Trimester of Pregnancy"

_animals, 2022, doi:10.3390/ani12111355_

Round 1
Reviewer 1 Report
This is an interesting paper, but it still, needs a significant work before being approved for publication. Please take into the following recommendations and questions.
Introduction:
Include information and references about Cortisol, Globulin A and TNFalpha as stress markers in saliva.
Line 65. Punctuation and space
Materials and methods:
Line 70. Please explain why you are using animals in a 3 weeks interval (12-15 weeks) to run your experiment. It means that there were some animals evaluated one week before parturition. So, how did you collect the 4-weeks samples?
Table 1. Please include metabolizable energy (Kcal/kg)
Line 89. How often the bar was changed or the essence was changed?
I would recommend the authors to include the post-parturition performance of the sows in terms of the weaning rate and litter weight at weaning and piglet average weight at weaning.
Results:
Line 178. TNF-α separate from in. It is observed through the text the α is not separated from the next word.
Table 4: Lower case letters in the same row indicate significant differences. How do interpret in row Birth weight (and others) ab vs ab vs a: Are they different or not? Please explain or fix.
Discussion
Line 217. Please explain why you use the adverb "probably" in this sentence.
Line 229-231. I don't understand what the authors want to say here and how it is related to the results obtained in this work.
Line 317-318. Elaborate the sentence "physiological stress is affected by the reproductive stage..."
Line 320. What does the word "above" refer to? The references must be written in the text based on the order number in the REFERENCES part (no author names mentioned in the text).
Line 321. Remove "will" and change increase to increases.
Line 322. Please elaborate or remove the phrase "the enrichment...conditions"
Line 338-344. Rewrite the whole paragraph. Support what you are saying: from one side you said enrichment does not affect reproduction but table 4 could be saying something different.
Author Response
Response to Reviewer 1 Comments
Point 1: (1) Line 65:Punctuation and space. (2) Line 178: TNF-α separate from in. It is observed through the text the α is not separated from the next word. (3) Line 321:Remove "will" and change increase to increases.(4) Line 217:Please explain why you use the adverb "probably" in this sentence.
Include information and references about Cortisol, Globulin A and TNFalpha as stress markers in saliva in introduction.
Response 1: Thank you very much for your valuable advice. I'm sorry for the above problems caused by my carelessness. We agree with you very much and have made corresponding modifications in the article. Studies have shown that the stereotypical behaviors mentioned in the article do harm to animal welfare, so the word "probably" is a clerical oversight that I have deleted.
Point 2: Materials and methods: (1) Line 70. Please explain why you are using animals in a 3 weeks interval (12-15 weeks) to run your experiment. It means that there were some animals evaluated one week before parturition. So, how did you collect the 4-weeks samples? (2) Table 1. Please include metabolizable energy (Kcal/kg). (3) Line 89. How often the bar was changed or the essence was changed? (4) I would recommend the authors to include the post-parturition performance of the sows in terms of the weaning rate and litter weight at weaning and piglet average weight at weaning.
Response 2:We very much agree with your questions.(1)Instead of three weeks apart, the data were collected continuously at 12.13.14.15 weeks, because studies have shown that the second trimester refers to 12 to 15 weeks. In the material approach we document how we collect behavioral, physiological data, right. You can look at the material method(parts 2.3 and 2.4) in detail. (2) We searched for these ingredients on the feed packaging bags fed by the pig farm at that time. But there was no information related to metabolizable energy, so we could not provide this information. (3) We did not replace the enrichment items during the test. (4) It's a pity, we didn't collect that data at the time, so we couldn't provide that information.
Point 3: Results:Table 4: Lower case letters in the same row indicate significant differences. How do interpret in row Birth weight (and others) ab vs ab vs a: Are they different or not? Please explain or fix.
Response 3:The letters on the same line are indistinguishable from each other, that is, the data are indistinguishable from each other. We are very sorry that we did not unify the data when we analyzed it, which led to the confusion of the results presented. Thank you very much for your question on this point, and we have revised it.
Point 4: Discussion (1) Line 229-231. I don't understand what the authors want to say here and how it is related to the results obtained in this work. (2) Line 322. Please elaborate or remove the phrase "the enrichment...conditions" (3) Line 320. What does the word "above" refer to? The references must be written in the text based on the order number in the references part (no author names mentioned in the text). (4) Line 317-318. Elaborate the sentence "physiological stress is affected by the reproductive stage..." (5) Line 338-344. Rewrite the whole paragraph. Support what you are saying: from one side you said enrichment does not affect reproduction but table 4 could be saying something different.
Response 4:After careful reading, we found that the expression was indeed vague and not very relevant to what the article said. We agree with your suggestions (1) and (2), which have been deleted in the article.(3)I want to express that the change of IgA is consistent with the result of the 12th reference mentioned above. In order to avoid being wordy, I just wrote above. We have adjusted it according to your opinion. (4)Cortisol is a product of the hypothalamic-pituitary-adrenal (HPA) axis, where activation of one affects the function of the other. Studies have shown that maternal HPA axis response to stress may vary with pregnancy. So physiological stress is also affected by the reproductive stage. (5)We are very sorry that you have such a question due to our carelessness. However, after our re-analysis and discussion, there is still no difference in the data in Table 4, and the results described in this paragraph are consistent with those presented in Table 4.
Thank you again for your patient guidance. Every problem you point out is very valuable to us. All of the above modifications are reflected in the article (marked in red). If there are still unclear explanations, please raise them. We are glad to receive your comments and revise them again.

Reviewer 2 Report
Overall this paper is interesting and conducted in a rigorous way. I have say though that it doesn't add an awful lot to the literature, which is rather expansive in this area.
There is a need to proofread this manuscript, although the general standard of English is high,
(L34) It is important to identify that sow stalls are not used in every jurisdiction. Bans have been implemented because they are know to significantly reduce welfare.
Likewise, I think the paper needs to work from this perspective. You are not "improving welfare" rather you are "reducing suffering". This may seem to be the same thing, but "improving welfare" has an implicit assumption that the animal is already in an acceptable state.
L21: three equal treatment groups (n=10)
What is Scent wood? How were these added? As shavings I assume.
L25: sham-chewing was
L26: How was preference measured?
L31: "a reduction in suffering" (or if preferred "welfare compromise")
L44 the EU has also banned sow stalls beyond the firsat 4 weeks of gestation, with some countries banning it altogether.
L47: "and promote"
L48: what is the "?"
L49: what is "late biting behaviour"?
L51: "on the behaviour and corticosteroid levels of..."
L55: high vs low stocking-density I assume?
L65: reducing welfare compromise
L61: Why only in the 3rd Trimester? Why not throughout pregnancy?
L74: I don't know what "full price" means here
L73: describe the wood, their composition, presentation and origin/manufacturer
L101: Instantaneous scan sampling
L103-4: I'm not sure what this means. It seems to be written incorrectly.
L106: Based on instantaneous sampling I don't think you need write this, in fact it is rather confusing as stated.
I would say that for so few state behaviours 1 observation every 5 minutes over 4h is quite short. Especially since it is from a recording
106-110: This is also confusing.
Table 2: In "social behaviour" what does "arching" mean?
Table 3: why "product" description? I would just say description
Please describe the "Duncan method"
Results: Please delete the journal guidance L144-
L162: at this point I am still unsure as to why you used two different pieces of wood (if I'm correct: one that was pine and one that smelled of pine). On what basis were you expecting a difference, was one more valuable than the other?
Murtagh et al., (2020) looked at scent/toy in dogs and similarly showed no effect of "biological salience". This may be useful in your discussion.
210-227 seems to be more introduction to the topic than a discussion of the results. I would suggest moving it.
229: frequency, not number
L232: but you weren't assessing social behaviours.. so this doesn't seem relevant (especially as it suggests that the sows were being kept in groups, not in stalls
L234: I would stress that this is a provision of the bare minimum (something is always better than nothing at all), but I don't think this could be considered "enough"
L242: "welfare compromise was reduced"
L243: Do you have evidence that it was more helpful in protecting the pighlets (e.g. number surviving to weaning). If not I would suggest "may be more helpful"
235: reduce, not improve.
256: Not sure re "limited in size ....fence" reference. I assume this is for pigs in your farms specifically?
4.2: this information about enrichment choice should be either in the intro or in the methods (around justification of object type)
Again 286-309 is rather lengthy as a discussion point and reads more like intro material.
L334: I think "fully express" is inappropriate here "express more of their innate behaviours" maybe
Author Response
Response to Reviewer 2 Comments
Point 1: (1) L25:sham-chewing was. (2) L47:"and promote". (3) L101:Instantaneous scan sampling. (4) Table 3: why "product" description? I would just say description. (5) Results: Please delete the journal guidance L144-. (6) L229: frequency, not number. (7) L235: reduce, not improve. (8) L232: but you weren't assessing social behaviours.. so this doesn't seem relevant (especially as it suggests that the sows were being kept in groups, not in stalls. (9) L210-227 seems to be more introduction to the topic than a discussion of the results. I would suggest moving it. (10) L51: "on the behaviour and corticosteroid levels of...". (11) L21: three equal treatment groups (n=10). (12) L334: I think "fully express" is inappropriate here "express more of their innate behaviours" maybe. (13) L234: I would stress that this is a provision of the bare minimum (something is always better than nothing at all), but I don't think this could be considered "enough".(14)L243: Do you have evidence that it was more helpful in protecting the pighlets (e.g. number surviving to weaning). If not I would suggest "may be more helpful". (15)L286-309 is rather lengthy as a discussion point and reads more like intro material. (16) 4.2: this information about enrichment choice should be either in the intro or in the methods (around justification of object type)
Response 1: Thank you very much for your valuable advice. I'm sorry for the above problems caused by my carelessness. We agree with you very much and have made corresponding modifications in the article.
Point 2: (1) What is Scent wood? How were these added? As shavings I assume. (2) L26: How was preference measured? (3) L48: what is the "?" (4) L49: what is "late biting behaviour"? (5) L55: high vs low stocking-density I assume? (6) L74: I don't know what "full price" means here. (7) Table 2: In "social behaviour" what does "arching" mean?
Response 2: (1)We use wood instead of shavings. We directly punch holes in wood and fix them to the limit rail. See 2.2 and Figure 2 for the production and hanging method of scented wood.
(2)Determine which toy the sow prefers by measuring how often she manipulates the toy, We have detailed records in section 3.3 of the results(3)Using EndNote to insert the literature automatically generated, we have modified.(4)I am very sorry for your incomprehension due to carelessness. Actually, tail biting.(5)Indeed, as you assume, it is the high and Low stocking-density. I have made appropriate modifications regarding the citation of this document (see Reference 36). (6)full-price granule compound feed. (7)Arching refers to the constant arching of a pig's nose or head due to discomfort
Point 3: (1) L34:It is important to identify that sow stalls are not used in every jurisdiction. Bans have been implemented because they are know to significantly reduce welfare. Likewise, I think the paper needs to work from this perspective. You are not "improving welfare" rather you are "reducing suffering". This may seem to be the same thing, but "improving welfare" has an implicit assumption that the animal is already in an acceptable state. (2) L31: "a reduction in suffering" (or if preferred "welfare compromise"). L65: reducing welfare compromise . L242: "welfare compromise was reduced"
(3) L44 the EU has also banned sow stalls beyond the firsat 4 weeks of gestation, with some countries banning it altogether.
Response 3: At present, there is no legal restriction on the use of sow limit pens in China. In addition, limit pens occupy a small area, improve management level, avoid fighting and reduce the probability of mechanical abortion, so some pig farms still use limit pens. However, the lack of movement of sows in the limit bar can lead to stereotypical behavior, so we quite agree with you that we are not improving welfare, but reducing suffering caused to sows in the limit bar. In view of your point of view, we have made adjustments in the article. Thank you very much for your more professional answer
Point 4: Why only in the 3rd Trimester? Why not throughout pregnancy?
Response 4: The third trimester is a relatively important period of pregnancy, because the growth pattern of the fetus during pregnancy is slow in the first trimester, fast in the second trimester, and faster in the third trimester, forming 1/3 of the fetal body weight in the third trimester.(L69-77)
Point 5: L73: describe the wood, their composition, presentation and origin/manufacturer
Response 5: composition :Agar, Pine essence and pigment. Presentation: Figure 2
Manufacturer: Foshan Songfeng Trading Co., LTD(Guang Dong, China)
Point 6: (1) L103-4: I'm not sure what this means. It seems to be written incorrectly. (2) L106: Based on instantaneous sampling I don't think you need write this, in fact it is rather confusing as stated.
I would say that for so few state behaviours 1 observation every 5 minutes over 4h is quite short. Especially since it is from a recording. (3) L106-110: This is also confusing.
Response 6: We very much agree with your views and doubts. After referring to the references, we have restated section 2.3. You can see it in the article. If you still have any questions, please point them out in time. We are happy to revise them.
Point 7: Please describe the "Duncan method"
Response 7: The data were processed with EXCEL 2016 and SPSS 22.0 software. GraphPad Prism8.0 was used for graphing. The difference analysis between the two groups was evaluated by paired T-test, while the comparison between the three groups was conducted by one-way ANOVA and Duncan method for multiple comparison.
Point 8: L162: at this point I am still unsure as to why you used two different pieces of wood (if I'm correct: one that was pine and one that smelled of pine). On what basis were you expecting a difference, was one more valuable than the other?
Murtagh et al., (2020) looked at scent/toy in dogs and similarly showed no effect of "biological salience". This may be useful in your discussion.
Response 8: Studies have shown that the use of aromatized objects reduced the duration of agonistic behaviors of the piglets and the exchange of fragrances increased the interest of the animals in the new object, reinforcing the notion that piglets alter their behaviors according to the environmental stimuli they receive. Scent wood is cheaper to make than pine, and the smell can last longer, so we wanted to compare whether scent wood could achieve the same effect as pine, thereby reducing the cost of animal suffering at the same time
Point9: 256: Not sure re "limited in size ....fence" reference. I assume this is for pigs in your farms specifically?
Response 9: At present, the size of the limit bar adopted in China is basically the same, and its space is basically the same as the body of a sow. The sow cannot adjust the direction of body, so the size and placement of toys are limited.
Thank you again for your patient guidance. Every problem you point out is very valuable to us. All of the above modifications are reflected in the article (marked in red). If there are still unclear explanations, please raise them. We are glad to receive your comments and revise them again.

Reviewer 3 Report
It is great that the authors tested for the effects of novel enrichment during pregnancy, which is a time when Sows natural behaviors are extremely limited when they are kept in farrowing crates. However, the authors should improve on the manuscript if they would like it to have a better impact on research. To do this, they should correct typos and grammatical errors throughout (several examples given below). They also could do a better job of citing other relevant literature on enrichment to sows during pregnancy (e.g. jute sacks). And I am especially concerned about their methods which are confusing and incomplete. The way that the behavioral data was coded and analyzed is not clear and needs to be clarified. In addition, full laboratory validations of the cortisol measurements (coefficients of variation, parallelism and reproducibility) are missing and should be presented.
Below are more specific comments:
Introduction
Lines 34-36 and throughout- choose either past or present tense, but do not switch between tenses (e.g. improves the utilization… reduces individual production costs).
Line 47- … and promote their reproductive performance
Line 48- typo (Telk?) and grammar (proved, not ‘had proved’).
Line 56- add reference.
Lines 60-62- is there any research on the use of enrichment for sows at any point in pregnancy? E.g. during the first or second trimesters? If so this would be relevant to review. I also find it hard to believe that there is no research on enrichment during the third trimester. For example, several studies have looked at the use of Jute sacks as ‘nesting material’ for sows, and I assume many of these studies looked into the third trimester. So I believe that this statement is false. What exactly is meant by ‘enrichment’ here should be clarified and the relevant literature should be more thoroughly reviewed.
Lines 62-63- how were these specific enrichment items chosen? What was the rationale for using them?
Materials and Methods
Lines 70-73- how was the sample size chosen? Was a power analysis run to determine the necessary sample size to determine a significant effect with 80% probability? How were Sows housed? In the same room? In different rooms? In adjacent stalls?
Figure 1 text is quite small and difficult to read. If the text is enlarged this would improve legibility.
Line 97- all activities of the sows
Lines 99-101- I am not familiar with the ‘behavior incidence rate’ and it is not clear what is meant here.
Lines 103-105- how was sham chewing identified using only a static screen? I assume that some video would need to be watched to determine whether the pigs were sham chewing or not at the given scan interval? Please clarify this.
Lines 106-107- what is meant by ‘within 5 minutes’? It is not clear how instantaneous sampling from a static screen and a 5-minute interval can be combined? This section should be written more clearly. What would happen if the sow exhibited two different behavioral states within the 5-minute interval? Would both be recorded?
Author Response
Response to Reviewer 3 Comments
Point 1: (1) Lines 34-36 and throughout- choose either past or present tense, but do not switch between tenses (e.g. improves the utilization… reduces individual production costs). (2) Line 47- … and promote their reproductive performance. (3) Line 56- add reference.(4) Line 217:Please explain why you use the adverb "probably" in this sentence. (5) Line 48- typo (Telk?) and grammar (proved, not ‘had proved’). (6) Line 97- all activities of the sows.
Response 1: Thank you very much for your valuable advice. I'm sorry for the above problems caused by my carelessness. We agree with you very much and have made corresponding modifications in the article.
Line217:Studies have shown that the stereotypical behaviors mentioned in the article do harm to animal welfare, so the word "probably" is a clerical oversight that I have deleted.
Line48-Telk? Because it is automatically generated when inserting the literature, I have made modifications
Point 2: Lines 60-62- is there any research on the use of enrichment for sows at any point in pregnancy? E.g. during the first or second trimesters? If so this would be relevant to review. I also find it hard to believe that there is no research on enrichment during the third trimester. For example, several studies have looked at the use of Jute sacks as ‘nesting material’ for sows, and I assume many of these studies looked into the third trimester. So I believe that this statement is false. What exactly is meant by ‘enrichment’ here should be clarified and the relevant literature should be more thoroughly reviewed.
Response 2: We quite agree with the question you have raised. So we carefully review the relevant literature, for sow welfare problems of a lot of research. But I am very sorry that in the relevant literature we have consulted, there is no relevant literature about welfare toys in the late or early and middle gestation. Maybe it is because my learning ability is not enough. If so, please kindly point it out and we will revise it again. Thank you very much for your valuable advice.
Point 3: Lines 62-63- how were these specific enrichment items chosen? What was the rationale for using them?
Response 3: Studies have shown that the use of aromatized objects reduced the duration of agonistic behaviors of the piglets and the exchange of fragrances increased the interest of the animals in the new object, reinforcing the notion that piglets alter their behaviors according to the environmental stimuli they receive. Scent wood is cheaper to make than pine, and the smell can last longer, so we wanted to compare whether scent wood could achieve the same effect as pine, thereby reducing the cost of animal suffering at the same time. We described it in the introduction(Line62-66)
Point 4: Figure 1 text is quite small and difficult to read. If the text is enlarged this would improve legibility.
Response 4: This image has a high resolution, so you can zoom in and still look sharp if you need
Point 5: Lines 70-73- how was the sample size chosen? Was a power analysis run to determine the necessary sample size to determine a significant effect with 80% probability? How were Sows housed? In the same room? In different rooms? In adjacent stalls?
Response 5: Your question is very worthy of our consideration and reference. But to be honest, we didn't do the analysis at the time. In order to exclude other factors, 30 French genetically pure large white pregnant sows (12 -15weeks of gestation) with the same genetic background, the third parity and expected delivery date (3 ± 1.5d) were randomly divided into 3 groups (n=10). During the trial, sows were kept in separate stalls, adjacent to each other, without contact, measuring 2.1m x 0.6m, with slatted floors, duck-billed drinking fountains and communal feeding troughs.
Point 6:Lines 99-101- I am not familiar with the ‘behavior incidence rate’ and it is not clear what is meant here.
Response 6: In order to avoid the influence of feeding time and hot temperature, we selected four time periods (7:00-8:00, 9:00-10:00, 14:00-15:00 and 16:00-17:00) for video observation and re-cording every 5 minutes completing 48 events per day (240 minutes of video recording per day). A frequency chart of postural behavior was established by recording the number of postural behavior events in each pig in 48 observed events per day to calculate the percentage of postural behavior per pig per day throughout the trial period.
Point 7: Lines 103-105- how was sham chewing identified using only a static screen? I assume that some video would need to be watched to determine whether the pigs were sham chewing or not at the given scan interval? Please clarify this.
Response7: Your question is very valid. I'm really sorry for your confusion due to our unclear expression in the behavior record (section 2.3). We did observe whether sows chew empty through video. It has been redescribed.
Point 8: Lines 106-107- what is meant by ‘within 5 minutes’? It is not clear how instantaneous sampling from a static screen and a 5-minute interval can be combined? This section should be written more clearly. What would happen if the sow exhibited two different behavioral states within the 5-minute interval? Would both be recorded?
Response 8: Your questions are very thoughtful and clearly point out the problem. According to your comments, we have restated it:During the experiment, all activities of the sows were filmed and documented from 6:00 am to 18:00 pm by digital video system ((Hikvision DS-ITS, Hangzhou Hikvision Co., Ltd., China). In order to avoid the influence of feeding time and hot temperature, we select-ed four time periods (7:00-8:00, 9:00-10:00, 14:00-15:00 and 16:00-17:00) for video observa-tion and recording every 5 minutes completing 48 events per day (240 minutes of video re-cording per day). In each event, each pig showed only one behavioral posture, and the postural behavior was recorded for a longer period of time if two different behaviors oc-curred. For other behaviors (sham-chewing, drinking, manipulation), the number and du-ration of each event were recorded. A frequency chart of postural behavior was established by recording the number of postural behavior events in each pig in 48 observed events per day to calculate the percentage of postural behavior per pig per day throughout the trial period.
Sows in each confinement stalls were individually assessed and the number of sows performing each behavior was recorded and transferred to a spreadsheet. The behavior de-termination method was defined in Table 2.
Thank you again for your patient guidance. Every problem you point out is very valuable to us. All of the above modifications are reflected in the article (marked in red). If there are still unclear explanations, please raise them. We are glad to receive your comments and revise them again.

Round 2
Reviewer 2 Report
I am happy that the authors have met my requirements.
Author Response
Thank you for your recognition of this article. Thanks to your guidance, our article has become more rigorous. Best wishes to you !
Reviewer 3 Report
Thank you for your efforts to revise the manuscript. It is improved but still very unclear in many parts. One thing that is needed is a thorough editing by an English native speaker. Without that many parts of the manuscript will be difficult to understand. For the methods, more clarity is needed especially on lines 126-127 where you state "In each event, each pig showed only one behavioral posture, and the postural behavior was recorded for a longer period of time if two different behaviors occurred." I still do not understand your methods for behavioral observation and this is critical for people to be able to understand. I think that you are trying to state that you recorded instantaneous data at 5 minute intervals, so every 5 minutes you recorded the posture of the pig, and their behavioral state. Is this correct? If you did something different, please clarify. But then what do you mean when you write "two different behaviors" above? And if it is an instantaneous scan sample, why would you lengthen the scan if 2 behaviors occurred? That does not make sense to me. You then state that for sham-chewing and manipulation you recorded the number and duration of each events, but for how long did you observe the pigs? Over 5 minutes X 48 = 240 minutes per sow? Or over a different time frame? It sounds like you combined instantaneous and continuous methods, but you need to clarify how you recorded each, and for what length of time you observed the behavior of each sow.
You have also not added your laboratory methods and validations. You need to report on how you processed the samples (any dilutions etc), how you ran them on the plate and at least very basic laboratory validations including parallelism and inter- and intra-assay coefficients of variation. Please check with your lab technicians and ask them to provide the basic quality checks of the analyses. Without these we have no idea if your physiological and immunological measurements worked.
Author Response
We are very sorry for the delay due to the language modification, and we are glad to receive your comments. Firstly, thank you very much for your detailed and patient guidance. We really recognize the questions you raised. For language problems, we have handed over to professional people to polish. After reading the question you raised, we found that there was indeed a big problem in the expression of behavior record. After discussion, we made the following modifications:
Point 1: For the methods, more clarity is needed especially on lines 126-127 where you state "In each event, each pig showed only one behavioral posture, and the postural behavior was recorded for a longer period of time if two different behaviors occurred." I still do not understand your methods for behavioral observation and this is critical for people to be able to understand. I think that you are trying to state that you recorded instantaneous data at 5 minute intervals, so every 5 minutes you recorded the posture of the pig, and their behavioral state. Is this correct? If you did something different, please clarify. But then what do you mean when you write "two different behaviors" above? And if it is an instantaneous scan sample, why would you lengthen the scan if 2 behaviors occurred? That does not make sense to me. You then state that for sham-chewing and manipulation you recorded the number and duration of each events, but for how long did you observe the pigs? Over 5 minutes X 48 = 240 minutes per sow? Or over a different time frame? It sounds like you combined instantaneous and continuous methods, but you need to clarify how you recorded each, and for what length of time you observed the behavior of each sow.
Response 1: During the experiment, all activities of the sows were filmed and documented from 6:00 am to 18:00 pm using a digital video system (Hikvision DS-ITS, Hangzhou Hikvision Co., Ltd., China). In order to avoid the influence of feeding time and hot temperature at noon, we selected four time periods (7:00-8:00, 9:00-10:00, 14:00-15:00 and 16:00-17:00) every day for observation and recording. Instantaneous scan sampling was performed for behavioral postures (standing, ventral lying, lateral lying, sitting), with recording being paused every 5 minutes and 48 events completed every day (240 minutes of video recording per day). A frequency chart of postural behaviors was established by recording the number of postural behavior events in each pig in 48 observed events per day to calculate the percentage of postural behaviors per pig per day throughout the trial period. For other behaviors (sham-chewing, drinking, manipulation), the frequency and time of occurrence were recorded during the observation period of 4 hours per day.
Sows in all confinement stalls were assessed individually and the number of sows performing each behavior was recorded and transferred to a spreadsheet. Behaviors are determined as defined in Table 2.
Point 2: You have also not added your laboratory methods and validations. You need to report on how you processed the samples (any dilutions etc), how you ran them on the plate and at least very basic laboratory validations including parallelism and inter- and intra-assay coefficients of variation. Please check with your lab technicians and ask them to provide the basic quality checks of the analyses. Without these we have no idea if your physiological and immunological measurements worked.
Response 2: Sow saliva was collected at 7:00-9:00 am every Wednesday for four weeks, using a self-made saliva collector (composed of medical gauze, medical degreased cotton, high-temperature sterilized disposable chopsticks and rubber bands), one sample per pig at a time. During sampling, the hand-held saliva collector was extended into the confinement stalls for the sow to chew freely. Once the cotton ball was wet, the saliva was collected in a centrifuge tube, followed by centrifugation for 10-15 min (3000 rpm) and storing the supernatant at -80 °C for further analysis. The ELISA kits (96T L180801256, L180710808, L190111127, WuHan) were used to determine the contents (ng/ml) of cortisol, immunoglobulin A, and tumor necrosis factor α in saliva as instructed by the manufacturer of the kit. Briefly, after the sample was diluted 1000 times (to determine the TNF-α index in the original solution), the capture antibodies to the cortisol, IgA, and TNF-α were coated to the wells of 96 ELISA plates, followed by incubating the samples with the kit solution and five washes in PBST to remove the nonspecific binding. Finally, the detection antibodies conjugated with horseradish peroxidase were incubated in the wells, and absorbance was measured at 450 nm using an ELISA reader. A standard curve was prepared with the concentration and absorbance of standards to produce a linear equation for quantifying the experimental samples. (Attachment 1)
Thank you for your generous guidance again, so that our article may look more rigorous. All of the above modifications are reflected in the article (marked in colour). If there are still unclear explanations, Please don't hesitate to point it out. We are glad to receive your comments and revise them again. Best wishes to you!